# Mechanically Induced Long-Period Fiber Gratings and Applications

Jiaqi Ran [1,2,3,†], Yarou Chen [1,2,3,†], Guanhua Wang [1,2,3,†], Zelan Zhong [1,2,3], Jiali Zhang [1,2,3], Ou Xu [1,2,3], Quandong Huang [1,2,3,*] and Xueqin Lei [4,5,*]

1   Institute of Advanced Photonics Technology, School of Information Engineering, Guangdong University of Technology, Guangzhou 511400, China
2   Key Laboratory of Photonic Technology for Integrated Sensing and Communication, Ministry of Education of China, Guangdong University of Technology, Guangzhou 511400, China
3   Guangdong Provincial Key Laboratory of Information Photonics Technology, Guangdong University of Technology, Guangzhou 511400, China
4   College of Physics and Electronic Information Engineering, Zhejiang Normal University, and Zhejiang Institute of Photoelectronics, Jinhua 321000, China
5   Department of Electrical Engineering, City University of Hong Kong, Hong Kong SAR, China
*   Correspondence: qdhuang@gdut.edu.cn (Q.H.); xueqilei@cityu.edu.hk (X.L.)
†   These authors contributed equally to this work.

**Abstract:** Long-period fiber gratings (LPFGs) functioning as band-reject filters have played a pivotal role in the realm of optical communication. Since their initial documentation in 1996, LPFGs have witnessed rapid advancements in areas such as optical sensing, the equalization of optical amplification, and optical band-pass filtering, etc. The unique attributes of optical fiber-based grating, including their miniaturized size, cost-effectiveness, and immunity to electromagnetic interference, have contributed significantly to various sectors over the last two decades. This paper presents a review of the evolution of LPFGs, with a specific focus on the progression and current trends of mechanically induced long-period fiber gratings. It offers a concise overview of coupled-mode theory, the fabrication processes, the merits, and the limitations associated with mechanically induced LPFGs. Moreover, this review elucidates the application methodologies of mechanically induced LPFGs and anticipates future directions in this field.

**Keywords:** long-period fiber grating; optical communication; fiber sensor





## 1. Introduction

Optical fiber grating sensors have garnered significant interest within the domain of optical communication, attributed to their myriad of benefits, which include high-resolution measurements, compactness, robustness, and the ability for long-range monitoring. Additional advantages are their ease of implementation for both point and quasi-distributed measurements, their adaptability to a wide variety of applications such as those in the oil and gas sectors or electromagnetic environments, and their cost-effectiveness. The intrigue in fiber gratings, which are characterized by a periodic modulation of the refractive index within the core or cladding of a fiber along its direction of propagation, has been widespread among scholars globally [1–20]. These fiber gratings are categorically divided based on their operational principles into two main types: fiber Bragg gratings (FBGs) and long-period fiber gratings (LPFGs). The distinction between them lies in the grating period; FBGs possess a grating period shorter than 1 μm, whereas LPFGs have a grating period exceeding 100 μm [1,21]. In the context of optical communication, LPFGs play a crucial role as band-reject filters. Nowadays, a principal application of such band-reject filters in fiber optics is embodied in the optical fiber sensor technology. This manuscript is dedicated to exploring the advancement of mechanically induced LPFGs, followed by a comprehensive review of their applications.

Despite the existence of multiple fabrication techniques for LPFGs, mechanically induced LPFGs possess distinctive and promising prospects of applications. The prevalent method for the fabrication of LPFGs involves the use of laser technology to periodically modify the fiber's refractive index, with options including ultraviolet (UV) laser, carbon dioxide ($CO_2$) laser, and femtosecond laser. Initially, UV laser-induced LPFGs are recognized for their high efficiency in inscription and good repeatability; however, the requirement for the use of photosensitive fiber and amplitude mask [21] along with the issue of degeneration have come to the fore. In contrast, $CO_2$ laser-induced LPFGs share these advantages without the requirement of photosensitive materials or amplitude mask, presenting a cost-effective and versatile solution [22,23]. However, femtosecond laser-induced LPFGs are hindered by the substantial costs associated with the equipment and its maintenance while holding promise in precision microfabrication [24]. As the development of fiber technology, such as the emergence of special fibers, continues to grow, the challenges encountered in laser-induced LPFG fabrication also become more prominent. The intricacies of microfabrication can intrude the fiber structure and potentially cause damage, which is particularly problematic for microstructure fibers, hollow-core fibers, anti-resonant fibers, and multi-core fibers. The Arc-induced LPFGs are recognized for their flexibility, stability, and cost-effectiveness [25]; however, their limited repeatability has restricted their utility. Additionally, reconfigurability is a paramount parameter in optical communication systems, where mechanically induced LPFGs offer comprehensive solutions. These mechanically induced LPFGs, which show the unique merits of reconfigurability, non-invasive modification, excellent flexibility, and minimal optical loss [26], are anticipated to play a significant role in future fiber-based systems, aligning with the evolving requirements of novel fiber technologies. Table 1 summarizes the main characteristics of several common long-period fiber grating fabrication techniques. Based on their respective features, these fabrication methods have different applications.

**Table 1.** Comparison of the LPFGs.

| | **Advantage and Disadvantage** |
|---|---|
| UV laser-induced LPFGs [21] | High efficiency in inscription, good repeatability, high requirement for photosensitive fiber and amplitude mask. |
| $CO_2$ laser-induced LPFGs [22,23] | Good repeatability, high efficiency in inscription, no requirement for photosensitive fiber and amplitude mask, and cost-effectiveness. |
| Femtosecond laser-induced LPFGs [24] | High efficiency in inscription, precision microfabrication, high costs of equipment and its maintenance. |
| Arc-induced LPFGs [25] | Good flexibility and stability, cost-effectiveness, limited repeatability. |
| Mechanically induced LPFGs [26] | Reconfigurability, non-invasive modification, excellent flexibility, and minimal optical loss. |

## 2. Fibers and Operation Principle of LPFGs

Long-period gratings (LPGs) are usually formed in the media of optical rectangular waveguides or optical fiber-based platforms. Optical rectangular waveguides, especially the low-refractive-index-contrast rectangular waveguides, have fiber compatibility merit and CMOS compatibility in fabrication. LPGs formed in rectangular waveguides achieve high performance in the mode conversion with ultra-broadband operation wavelength [27,28]. Optical fibers are a special form of the low refractive index contrast rectangular waveguides, where the core and cladding refractive index difference ranges from 0.004 to 0.02 in standard communication fiber cables. According to the fiber V-number, optical fibers are classified as single-mode fibers (SMFs) and multi-mode fibers (MMFs). The single-mode regime is observed when V < 2.405; otherwise, the regime is multi-modal. The unique propagation mode in the case of V < 2.405 is LP01. The few-mode fiber (FMF) is a type of optical fiber

with a core diameter that falls between those of single-mode and multi-mode fibers. Based on the main content of the article, this paper mainly introduces SMFs and FMFs. Schematic diagrams of the fibers and their mode field patterns are shown in Figures 1–4.

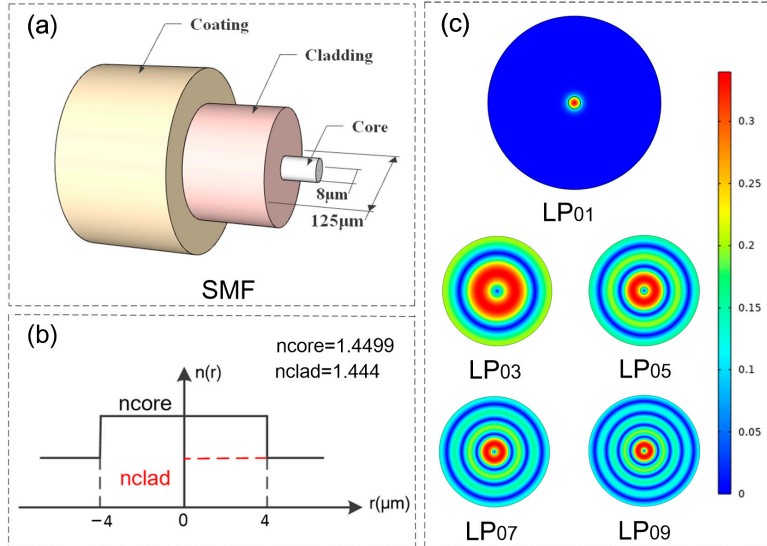

**Figure 1.** (**a**) Diagram, (**b**) refractive index profile, and (**c**) mode fields of single−mode fiber.

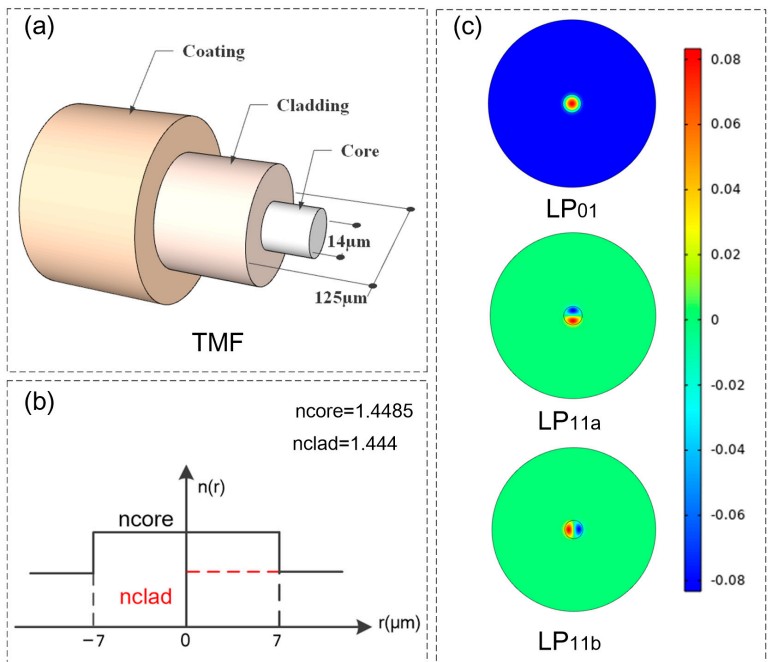

**Figure 2.** (**a**) Diagram, (**b**) refractive index profile, and (**c**) mode fields of two−mode fiber.

Traditional single-mode fiber is usually known as standard SMF-28. As shown in Figure 1, a diagram of the structure, index profile and mode fields of a single-mode fiber are demonstrated, which supports only one guided mode ($LP_{01}$ mode) and many leaky cladding modes. The core and cladding diameters of the single-mode fiber are 8 μm and 125 μm, and the refractive index difference is 0.0059. The $LP_{01}$ mode can excite a series of cladding modes, which are labeled as $LP_{03}$, $LP_{05}$, $LP_{07}$, and $LP_{09}$ cladding modes, where the cladding modes are of high loss. LPFG serves to couple the core mode to the cladding modes.

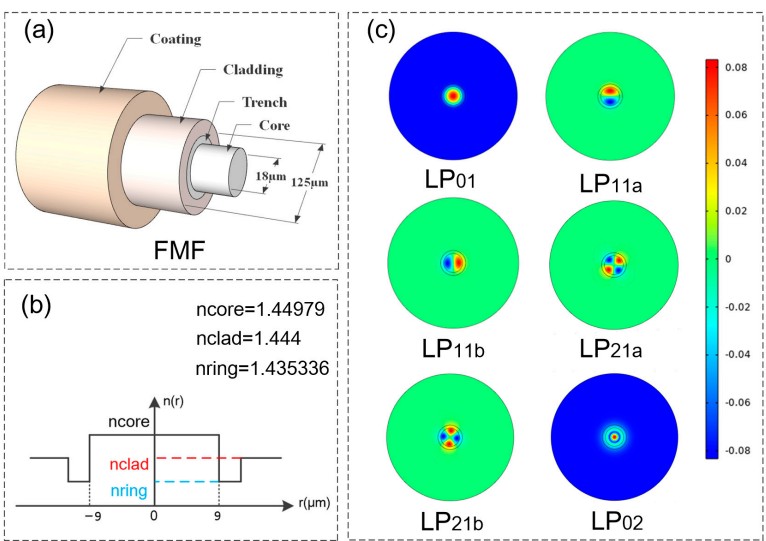

**Figure 3.** (**a**) Diagram, (**b**) refractive index profile, and (**c**) mode fields of four−mode ring core fiber.

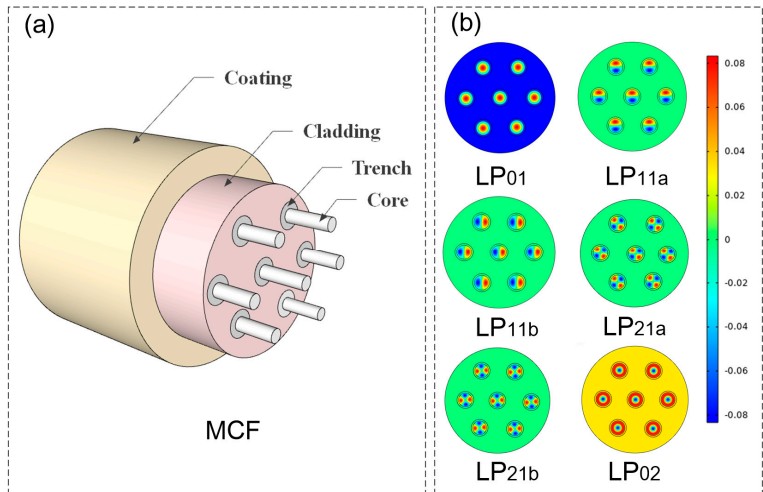

**Figure 4.** (**a**) Diagram and (**b**) mode fields of seven-core six-mode fiber.

When the core size enlarges, the fiber can support more core modes. A two-mode step-index fiber is shown in Figure 2, which supports the $LP_{01}$ and $LP_{11}$ modes. The core and cladding diameters of the two-mode fiber are 14 μm and 125 μm, and the refractive index difference is 0.0045. LPFG serves to couple one of the core modes to the other core mode. Such LPFGs are known as the mode converters.

To reduce the leaky loss of the guided modes due to the bending, a low refractive index ring core is introduced in the fiber design. Figure 3 shows a four-mode fiber with a ring core, where the core diameter is 18 μm; the ring has the inner-ring and outer-ring diameters of 18 μm and 22 μm, respectively; and the cladding diameter is 125 μm. However, there are some challenges to form the LPFGs via laser microfabrication. Mechanically induced LPFGs show unique merits in these kinds of fibers.

To increase the communication capacity, multi-core few-mode fiber has been proposed in recent years. Figure 4 illustrates a seven-core six-mode fiber, with each core accommodating a six-mode fiber core featuring a ring structure. The core diameter of the six-mode fiber is 26 μm, the ring has inner-ring and outer-ring diameters of 26 μm and 32 μm, and the cladding diameter is 200 μm. Mechanically induced LPFGs can manipulate the mode conversion for the seven cores simultaneously. By knowing the fiber parameters, LPFGs can be formed in different categories of fibers to manipulate any order of modes and to serve in different applications.

The analysis of LPFG is via the coupled-mode theory or transfer-matrix method. The coupled-mode theory is commonly employed to analyze the optical characteristics of uniform gratings. For long-period gratings with non-uniform periods, the transfer-matrix method is often used for study. This is because in non-uniform long-period gratings, the parameters such as grating coupling coefficients and propagation constants vary along the fiber axis, making the calculation process for mode coupling theory analysis more intricate. Additionally, the accuracy of mode coupling theory decreases when the refractive index modulation is significant.

It is well known that the resonant wavelength of LPFGs is expressed as

$$\lambda_{res} = \left( n_{eff\,1} - n_{eff\,2} \right) \Lambda, \tag{1}$$

where $\lambda_{res}$ is the resonant wavelength, $n_{eff\,1}$ is the effective index of the core mode or any order of the guided mode in a few-mode fiber, $n_{eff\,2}$ is the effective index of the cladding mode or any higher order mode in a few-mode fiber, and $\Lambda$ is the grating period. The coupling efficiency of the long-period grating depends on the perturbation to the fiber, which can be represented by the overlap integral between the two desired modes, expressed as

$$\kappa = \frac{\omega \varepsilon_0 \left( n_{eff\,1}^2 - n_{eff\,2}^2 \right)}{2\pi} \iint_s \vec{E_1} \cdot \vec{E_2} dS, \tag{2}$$

where $\omega$ is the angular optical frequency, $\varepsilon_0$ is the vacuum dielectric constant, $n_{eff\,1}^2$ and $n_{eff\,2}^2$ are the effective indices of the desired modes, and $\vec{E_1}$ and $\vec{E_2}$ are the normalized electric fields of the coupling modes, respectively. The resonant dips of LPFGs are controlled by the above parameters.

LPFGs can be formed through many techniques, such as laser-induced refractive index change [29–31], laser-induced structural change [32], and mechanically induced refractive index/structural change [33], etc. Among them, the catalog of the mechanically induced, for which the main principle is the refractive index modulation caused by the elastic–optical effect of optical fiber, can be mainly divided into two kinds. One type (Type (1)) is the mechanically induced refractive index change.

As shown in Figure 5a, long-period alloy waveguide grating (LPAWG) in a regular waveguide profile is predefined on one of the slabs, and fiber is clamped in the middle via another blank slab. The front view of the setup is shown in Figure 5b, where the grating period is defined as $\Lambda$. The refractive index of the fiber is periodically modulated when force is applied to the top of the slab. The advantage of long-period fiber gratings prepared by this structure is that the grating period can be flexibly controlled and shows low insertion loss, batch production, and reconfigurability. However, mechanical devices usually have higher requirements on environmental conditions and need to be carried out under vibration-free conditions, and LPFG prepared by this structure has the disadvantage of obvious degradation, and grating characteristics are prone to change over time. This shortcoming will limit its practical application. In addition, the drawback of this structure is that the required force is relatively large. The other type (Type (2)) is the mechanically induced structural change. As shown in Figure 6a, long-period alloy waveguide grating (LPAWG) with a triangle profile is predefined on both slabs, and fiber is clamped in the middle of the slabs. The front view of the setup is shown in Figure 6b, where the grating period is defined as $\Lambda$. One beneficial aspect of this design is the adaptability. By adjusting the angle between the fiber and the grooves, one can modify the grating period and hence the positions of the notches. Similarly, it is easily achievable by altering the length of fiber under pressure and adjusting the notch linewidths. Additionally, notch depth can be tuned by adjusting the pressure. Upon removal of the perturbation, the fiber's transmission returns to its initial state. Consequently, a diverse array of filter functions can be achieved by using the same grooved plate and fiber. Compared with the structure in Figure 5, the advantages of this structure include low required force, batch production, and

reconfigurability. The drawback of this structure is that the bending loss will occur, and as a result, unfortunately, the structure will suffer an exceptional insertion loss in some way.

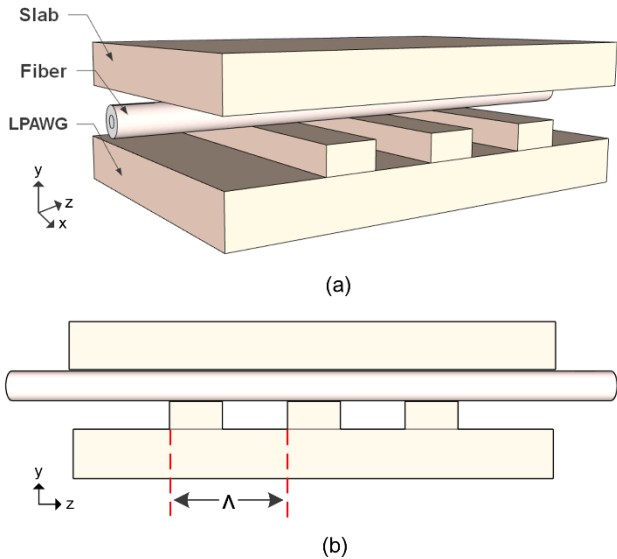

**Figure 5.** (**a**) Schematic diagram of Type (1) mechanically induced LPFG via mechanically induced refractive index change; (**b**) front view of the setup.

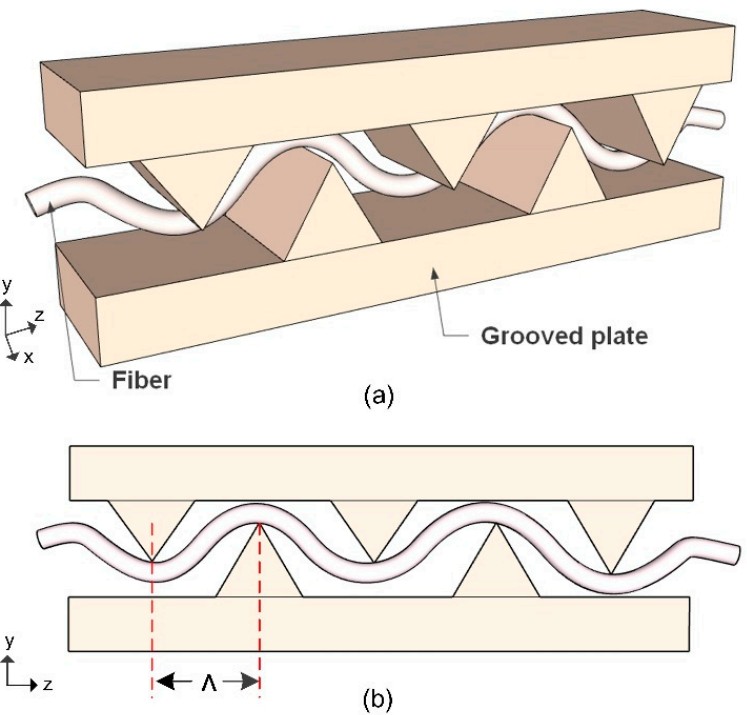

**Figure 6.** (**a**) Schematic diagram of Type (2) mechanically induced LPFG via mechanically induced structural change; (**b**) front view of the setup.

### 3. Applications of Mechanically Induced LPFGs

The mechanically induced LPFGs formed in the fiber possess numerous applications. Here is a summary of the applications of the mechanically induced LPFGs, shown in Figure 7. Obviously, fiber sensor is the main application based on the mechanically induced LPFG platform. Meanwhile, the mechanically induced LPFGs can also be used for optical signal processing, mode multiplexing, filter, interferometer, etc. In the following

sections, we will discuss some of the mechanically induced LPFGs in applications and their future trends.

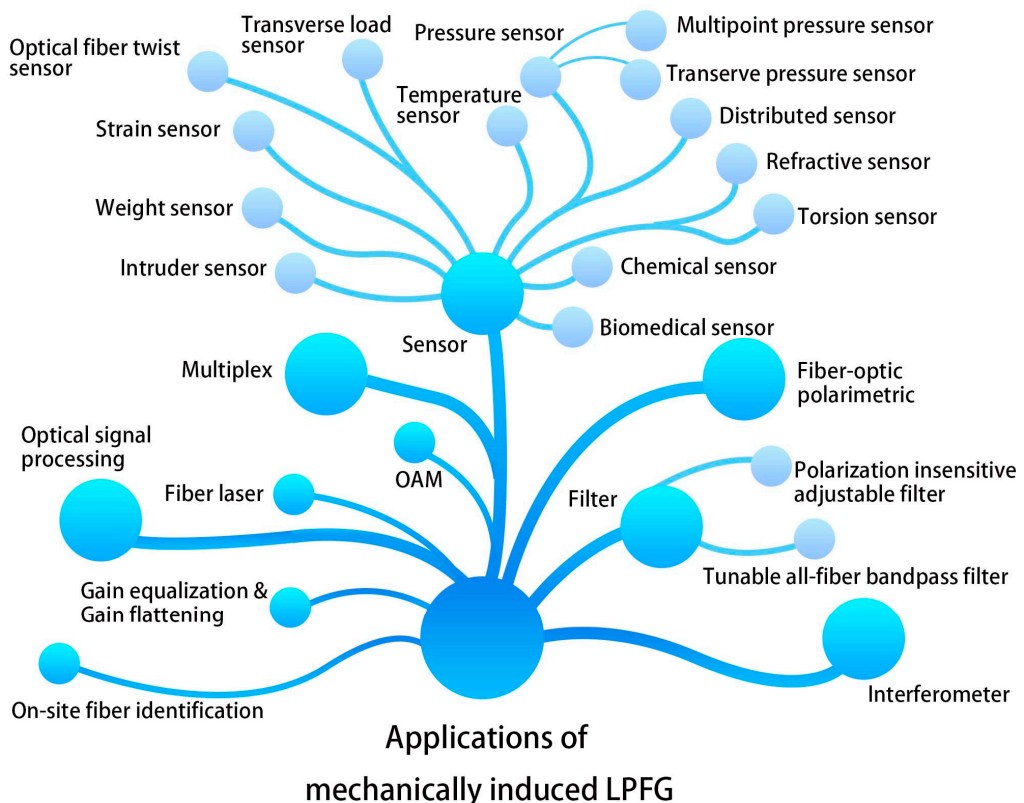

**Figure 7.** A summary of the applications of the mechanically induced LPFGs.

As mentioned in the previous introduction, two types of the mechanically induced LPFGs are used. In 1986, Blake et al. [34] utilized a mechanically induced method to squeeze a two-mode fiber and successfully converted the $LP_{01}$ mode into the $LP_{11}$ mode. From this perspective, the preparation of mechanically induced LPFGs preceded that of 1996 by approximately a decade [34]. Both types of the structure are widely used in the transmission as band rejectors [35–43]. Such band rejectors are further developed to serve as transverse load, temperature, pressure, twist, and refractive index sensors [44–59].

In early 2003, G. Rego et al. [36] proposed a novel technique for producing mechanically induced LPFGs. The fabrication of mechanically induced LPFGs involves winding a string around a fiber/grooved tube set. Tunability of resonant wavelengths is achieved through cladding etching before grating fabrication. Results demonstrate that the produced mechanically induced LPFGs exhibit low insertion loss (<0.2 dB) with bandwidths ranging from 5 to 30 nm and the isolation of loss peaks higher than 20 dB. Precise isolation control is achieved by adjusting fabrication parameters and string turns. External loads or tube diameter changes offer additional control options. These mechanically induced LPFGs show high sensitivity to loads (20 dB/kg), making them suitable for multi-parameter optical sensors. Additionally, the technique enables excellent control over the gratings' spectra, which is fundamental for achieving gain equalization in fiber amplifiers. Later, such mechanically induced LPFGs are widely used in the detection of transverse load, pressure, and temperature [45–47].

In 2010, Y. Jiang et al. [45] reported a work on a transverse pressure sensor based on mechanically induced LPFGs and fiber loop ring-down technique. The work focused on investigating the transmittance characteristics of mechanically induced LPFGs when spliced into a fiber loop, introducing an extra loss that affects the ring-down time. The results demonstrated that the difference between the reciprocals of the ring-down time

with and without pressure exponentially increased with increasing pressure in the range of 0–23.4 MPa. The device achieved high sensitivity in detecting pressure changes, and it can be used for applications such as pressure sensing in various fields.

In 2020, A. Das et al. [46] reported a reflectometric optical fiber sensing technology based on mechanically induced cascaded LPFG. They developed a reflectometric configuration of the cascaded LPFG structure by exploiting the Fresnel reflection at the end of sensing fiber, utilizing merely a single LPFG that is mechanically induced in the sensing fiber. Due to a photo-elastic effect, LPFG will be induced when the fiber is sandwiched between casings and pressed by tightening screws. This sensor can accurately measure loads up to 10 N without causing any permanent changes in the fiber properties. Furthermore, it exhibits strain and temperature sensitivities of 9.4 nm/% and 0.06 nm/$^\circ$C, respectively. The statement highlights its potential for pressure, strain, and temperature sensing applications. Additionally, when subjected to external physical processes, this sensor clearly exhibits shifts in attenuation bands, which can be utilized for non-destructive evaluations. In 2021, I. Torres-Gómez et al. [47] reported the simultaneous measurement of transverse load and temperature. The study involved the connection of a mechanically induced LPFG and an LPFG inscribed by a continuous-wave $CO_2$ laser. The mechanically induced LPFG was utilized for transverse load measurement, while the $CO_2$ laser-induced LPFG was employed for temperature measurement. Both transverse load and temperature measurements demonstrated high repeatability in both individual and simultaneous processes. Additionally, the multiplexed LPFGs exhibited low-cladding-mode crosstalk for transverse load and temperature. This technique has the potential to enhance the implementation of fiber optic-distributed sensing systems based on the wavelength division multiplexing of LPFGs.

The mechanically induced LPFGs also serve as temperature sensors [48–50], while the model of the LPFG can be formed by using 3D printing techniques in the fabrication process [51]. In 2011, L. A. García-de-la-Rosa et al. [52] reported the spectral response of a mechanically induced LPFG to the ambient temperature variation. The mechanically induced LPFG, created by using a pressure rig with aluminum grooved plates, exhibited significant wavelength shifts and contrast decrease in the attenuation bands with temperature changes. The study revealed that the wavelength shift curve presented a near-linear region from 0 to 40 $^\circ$C with a sensitivity of 180 pm/$^\circ$C and a flattened zone beyond this temperature. Meanwhile, the contrast of the attenuation bands rapidly decreased from the maximum value to 0 dB with a quasi-cosinoidal behavior. The mechanically induced LPFG can be used as low-cost ambient temperature sensor through intensity-based measurements. In 2021, T. Chinggungval et al. [53] explored the feasibility of utilizing mechanically induced LPFGs as optical fiber-based sensors. The grating device is pressed onto an optical fiber to observe changes in resonance wavelengths at different groove periods. Subsequently, an optical time domain reflectometer (OTDR) is employed to analyze light behavior, event positions, and losses when resonant wavelengths are known. The characteristic of the grating is identified as non-contradirectional coupling based on non-reflective event observations from the OTDR trace, and the attenuation of resonant wavelengths follows a cubic function in response to the applied force. This method shows great potential for sensing applications in vulnerable environments, serving as a low-cost device for various sensor applications, including intruder sensors and weight sensors. The mechanically induced LPFGs are widely used as twist sensors [54–59].

In 2018, M. Zhang et al. [60] proposed a fiber liquid level sensor based on a reflective mechanically induced LPFG. Operating as a Michelson interferometer, the sensor incorporates a mechanically induced LPFG with double-cladding fiber (DCF), where the end-face is coated with a silver film. DCF is employed to enhance the generation of interference spectra. Liquid levels are measured by monitoring the shift in interference fringes. Both theory and experiments illustrate certain relationships between the interference fringe spacing and the distance from the grating to the reflector. The fringe contrast ratio is influenced either by the stress or the position of the mechanically induced LPFG. Results

demonstrate that the sensor's sensitivity reaches 0.03136 nm/mm, surpassing conventional interferometric liquid level sensors. Furthermore, the device is cost-effective with a simple structure, showcasing significant potential for various other optical fiber sensing applications. The mechanically induced LPFGs are also widely used in applications such as tunable band-passed filter [61], ring laser in the L-band [62], EDFA gain flatter [63], OAM mode generator [64,65], and mode converter [66,67]. The applications of the mechanically induced LPFGs are advancing toward the future. In the future, the mechanically induced LPFGs can be developed further to be used in an optical communication system as mode converters in O band or 2 μm wavelengths, or in extreme areas, where LPFGs can be formed using lasers, or in special application areas such as in airspace or submarines. All in all, there will be a bright future for the mechanically induced LPFGs, as predicted. Table 2 summarizes the different applications and performance characteristics of two types of mechanically induced LPFGs.

**Table 2.** Some applications of the two types of the mechanically induced LPFGs.

| | Applications | Performance | Reference |
|---|---|---|---|
| Type (1) | Modal interferometer | Small device size, good compatibility. | [38] |
| | Band rejectors | Reconfigurable, erasable, high thermal stability. | [40–43] |
| | Transverse load/pressure sensor | Good repeatability, higher sensitivity. | [45,47] |
| | Strain and temperature sensor | Potentially enhanced degree of freedom in controlling temperature sensitivity, and a short length of sensing fiber. | [46,48–51] |
| | Refractive index sensors | Offering flexibility and real-time response. | [52,53] |
| | Tunable bandpass filter | Single-fiber configuration without insertion devices and independence between tunable range and transmission amplitude. | [61] |
| | Ring laser in the L-band | The controllable wavelength range was up to 40 nm. | [62] |
| | OAM mode generator | Obtaining a working bandwidth of 114 nm. | [64] |
| Type (2) | EDFA's gain equalization | Improving gain-flattening and enhancing the gain characteristics of the EDFA. | [36,63] |
| | Twist sensors | High temperature stability, high sensitivity of twist sensor. | [54–59] |
| | Liquid level sensors | High sensitivity. | [60] |
| | Mode converter | Reconfigurability. | [66,67] |

## 4. Conclusions

In conclusion, we have made a brief review of the mechanically induced LPFGs, and a summary of the applications. The mechanically induced LPFGs show unique merits of reconfigurability, repeatability, and low cost. In the future, the mechanically induced LPFGs will continue in the production stage and more application areas will appear. The mechanically induced LPFGs can be developed to be used in the optical communication system as mode converters in wider wavelengths, or in extreme areas where the formation of LPFGs can be satisfied by using lasers, or special application areas such as airspace or in submarines.

**Author Contributions:** Conceptualization, J.R. and Y.C.; methodology, J.R. and Y.C.; software, J.R. and Y.C.; validation, J.R., Y.C., G.W. and Z.Z.; formal analysis, J.R.; investigation, Q.H.; resources, Q.H.; data curation, J.R.; writing—original draft preparation, J.R.; writing—review and editing, J.Z., O.X., Q.H. and X.L.; visualization, Q.H. and X.L.; supervision, Q.H.; project administration, Q.H. All authors have read and agreed to the published version of the manuscript.

**Funding:** This research received no external funding.

**Institutional Review Board Statement:** Not applicable.

**Informed Consent Statement:** Not applicable.

**Data Availability Statement:** Data will be made available on request.

**Conflicts of Interest:** The authors declare no conflict of interest.

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
