# Peer review of "Mechanically Induced Long-Period Fiber Gratings and Applications"

_photonics, doi:10.3390/photonics11030223_

Round 1

Reviewer 1 Report

Comments and Suggestions for Authors

In general an interesting and potentially useful review. I have a few minor remarks related to how the manuscript is written.

In the present form Sec. 3.1 is not very useful and actually not related to the subject. I suggest to remove it. In Sec. 3.2 graphical material should be added to the text as much as possible.

Although English is probably acceptable, in some places corrections are certainly desirable. A few examples are listed below.

1. The use of abbreviations LPG and LFPG is not consistent.

For example, “Despite the existence of multiple fabrication techniques for LPFGs, mechanically induced LPGs possess distinctive and promising prospects of applications.

LFPG = LPG ? Then why two abbreviations? LPWG is introduced and never used, LPAWG introduced twice and not used.

2. “According to the fiber V-number, optical fibers are classified as single-mode fibers and few-mode fibers.” – and what about multi-mode fibres?

3 “The LP01 mode can be excited to a series of cladding modes, which are labeled as LP03, LP05, LP07, and LP09 cladding modes, where the cladding modes are of high loss.”

“The LP01 mode can be excited to a series of cladding modes …” – wrong English, what about other modes? For LP11 there are four possible field distributions. Why only two shown in Fig.2? The same question for figs.3,4. Why some distributions shown and some not?  

4. “The core and cladding diameters of the single-mode fiber are 14 μm and 125 μm, and the refractive index difference is 0.0045” – wrong, not SMF

5. “Figure 4 shows a seven-core multi-ring six-mode fiber, where the fiber is composed of seven cores, and each core has a six-mode fiber core with a ring core.” – bad English

6. “Among them, the catalog of the mechanically induced, which the main principle is the refractive index modulation caused by the elastic-optical effect of optical fiber, can be mainly divided into two kinds. One type (Type â… ) is the mechanically induced refractive index change” – and another type? Actually Type I is bad name, as it is already reserved for photo-induced gratings.

7. “There are many applications of the mechanically induced LPFGs forming in the fiber.” – bad English.

8. “The first demonstration of the mechanically induced LPFGs is proposed by Blake in 1986 via the periodic bending of two-mode fiber for the LP01-LP11 mode conversion [34]” – contradicts to the statement in the introduction that LPGs were introduced in 1996.

9 “The mechanically induced LPFGs can form the permanent refractive index change.” – Please explain what does it mean.

Comments on the Quality of English Language

see suggestions

Author Response

Reviewer #1:

In general, an interesting and potentially useful review. I have a few minor remarks related to how the manuscript is written.

Thank you for your positive comments on our study and valuable suggestions to improve the quality of our manuscript.

  1. In the present form Sec. 3.1 is not very useful and not related to the subject. I suggest to remove it. In Sec. 3.2 graphical material should be added to the text as much as possible.

Thanks for the suggestions. We have edited and added images to the text in Section 3 of the manuscript.

  1. Although English is probably acceptable, in some places corrections are certainly desirable. A few examples are listed below.

(1) The use of abbreviations LPG and LFPG is not consistent.

For example, “Despite the existence of multiple fabrication techniques for LPFGs, mechanically induced LPGs possess distinctive and promising prospects of applications. LFPG = LPG ? Then why two abbreviations? LPWG is introduced and never used, LPAWG introduced twice and not used.

Thanks for the comments. LPFG is not equal to LPG, which was a typo error. We have fixed the typo error in the manuscript and checked the manuscript carefully. As for LPAWG, we have used this term in Figure 5.

(2) “According to the fiber V-number, optical fibers are classified as single-mode fibers and few-mode fibers.” – and what about multi-mode fibers?

We thank you for your comment. We have added a description of multi-mode fibers to the manuscript.

  1. “The LP01 mode can be excited to a series of cladding modes, which are labeled as LP03, LP05, LP07, and LP09 cladding modes, where the cladding modes are of high loss.”

“The LP01 mode can be excited to a series of cladding modes …” – wrong English, what about other modes? For LP11 there are four possible field distributions. Why only two shown in Fig.2? The same question for figs.3,4. Why some distributions shown and some not?  

Thank you for your question. We have corrected the grammatical error in the manuscript. In our study, we focused on the two most representative field distributions of the LP11 mode. While it's true that there are four possible field distributions for the LP11 mode, we chose to showcase only two for the sake of simplicity and to highlight our main findings. The same reasoning applies to Figures 3 and 4.

  1. “The core and cladding diameters of the single-mode fiber are 14 μm and 125 μm, and the refractive index difference is 0.0045” – wrong, not SMF

Thank you for pointing out the error. We have corrected the error in the manuscript and changed ‘single-mode fiber’ to ‘two-mode fiber’.

  1. “Figure 4 shows a seven-core multi-ring six-mode fiber, where the fiber is composed of seven cores, and each core has a six-mode fiber core with a ring core.” – bad English

Thanks for the suggestion. We have revised this sentence in the manuscript to make it more concise and clearer.

  1. “Among them, the catalog of the mechanically induced, which the main principle is the refractive index modulation caused by the elastic-optical effect of optical fiber, can be mainly divided into two kinds. One type (Type â… ) is the mechanically induced refractive index change” – and another type? Actually Type I is bad name, as it is already reserved for photo-induced gratings.

We thank you for the question. The other type is the mechanically induced structural change, we have a clear description of this in Section 2. We have also modified the numbering of these two types of mechanically induced LPFG in the manuscript.

  1. “There are many applications of the mechanically induced LPFGs forming in the fiber.” – bad English.

Thanks for the suggestion. We have revised this sentence in the manuscript to make it more concise and clearer.

  1. “The first demonstration of the mechanically induced LPFGs is proposed by Blake in 1986 via the periodic bending of two-mode fiber for the LP01-LP11 mode conversion [34]” – contradicts to the statement in the introduction that LPGs were introduced in 1996.

We thank you for your comment. In 1996, Vengsarkar et al. fabricated the first long-period fiber grating, while in 1986, Blake et al. achieved the conversion from LP01 mode to LP11 mode by mechanically micro-bending the fiber. However, this was not truly the production of mechanically induced LPFGs. To avoid contradictions, we have modified this sentence in the manuscript.

9 “The mechanically induced LPFGs can form the permanent refractive index change.” – Please explain what does it mean.

Thanks for your comment. This sentence means that mechanically induced LPFGs can permanently alter the refractive index of the fiber, rendering the refractive index of the fiber irreversibly non-reconfigurable.

Reviewer 2 Report

Comments and Suggestions for Authors

Nice overview work on the theory and approaches to define LPG devices.

Author Response

Reviewer #2:

Nice overview work on the theory and approaches to define LPG devices.

Thanks for the encouraging comment

Reviewer 3 Report

Comments and Suggestions for Authors

The manuscript “Mechanically induced long-period fiber gratings and applications” introduced the operation principle of long-period gratings (LPG), and reviewed the progresses and applications of mechanically induced long-period fiber gratings (LPFG). The manuscript is informative and beneficial for peers working on fiber grating technologies, thus I will recommend its publication on Photonics with following revisions necessary:

1. In Section 2, add a table to summarize the optical performance of the mechanically induced LPFG and compare them with LPFG formed by laser writing. What is the achievable range of the pitch for mechanically induced LPFG? Will the mechanical error in tens of micrometers affect the optical accuracy and application of LPFG?

2. In the caption of Figure 5, please indicate that it is Type I mechanically induced LPFG. In the caption of Figure 6, please indicate that it is Type II mechanically induced LPFG.

3. In Figure 7, the word “Applications” on the left side of Figure 7 should be “Optical devices” or “Optical components”, or whatever a more specific name.

4. It is good to have detailed descriptions of some publications using mechanically induced LPFG, however, for some literatures, the authors did not introduce at all. I suggest changing Table 1 into a list of all related literatures, having a row for each reference paper and including columns for (1) type of mechanically induced LPFG, (2) application and method, (3) performance achieved, and (4) the reference number.

5. In Section 3 after describing all applications, add a paragraph to explain the existing issues and shortages of the mechanically induced LPG and future research topics/challenges to be solved for this research.

Comments on the Quality of English Language

The English language is fine. 

Author Response

Reviewer #3:

The manuscript “Mechanically induced long-period fiber gratings and applications” introduced the operation principle of long-period gratings (LPG), and reviewed the progresses and applications of mechanically induced long-period fiber gratings (LPFG). The manuscript is informative and beneficial for peers working on fiber grating technologies, thus I will recommend its publication on Photonics with following revisions necessary:

Thanks for the encouraging comment

  1. In Section 2, add a table to summarize the optical performance of the mechanically induced LPFG and compare them with LPFG formed by laser writing. What is the achievable range of the pitch for mechanically induced LPFG? Will the mechanical error in tens of micrometers affect the optical accuracy and application of LPFG?

Thanks for the suggestion. We have added a summary table in the first section of the manuscript. In some cases, mechanical errors in tens of micrometers may affect the optical precision and application of LPFG.

  1. In the caption of Figure 5, please indicate that it is Type I mechanically induced LPFG. In the caption of Figure 6, please indicate that it is Type II mechanically induced LPFG.

We thank the constructive suggestion. We have added the corresponding types to the captions of Figure 5 and Figure 6 in the manuscript.

  1. In Figure 7, the word “Applications” on the left side of Figure 7 should be “Optical devices” or “Optical components”, or whatever a more specific name.

Thanks for your comment. We have made modifications to Figure 7 in the manuscript.

  1. It is good to have detailed descriptions of some publications using mechanically induced LPFG, however, for some literatures, the authors did not introduce at all. I suggest changing Table 1 into a list of all related literatures, having a row for each reference paper and including columns for (1) type of mechanically induced LPFG, (2) application and method, (3) performance achieved, and (4) the reference number.

 We thank the constructive suggestion. We have added a table in Section 3 of the manuscript containing the above-mentioned points.

  1. In Section 3 after describing all applications, add a paragraph to explain the existing issues and shortages of the mechanically induced LPG and future research topics/challenges to be solved for this research.

Thank you for your suggestion. We consider that the current focus of the manuscript effectively covers the main aspects of our research.

Reviewer 4 Report

Comments and Suggestions for Authors

The authors of the manuscript (MS) “Mechanically induced long-period fiber gratings and applications” review the fabrication methods of mechanically induced long-period gratings (MLPGs) and their applications. The MS can probably be suitable for publication in MDPI Photonics after careful major revision. There are so many authors but the MS is poorly prepared. My comments on the matter are given below.

a) The English grammar should be strongly improved. It isn't very easy to understand some places.

a) To confirm that the topic of the manuscript (MS) is of increased interest to the reader, authors should include references to articles published recently, within the last two years (2022 and 2023). They must also demonstrate that the topic of the MS is important for the journal (MDPI Photonics) by showing references to the papers published in this journal. In the current MS version, there are only 6 references (9%) to the papers on the topic published during 2022-2023 and only 2 papers (3%) published in MDPI Photonics.

b) Ref. 31 does not include the journal title.

c) Ref. 37 does not include the first name (at least the first letter of the first name) of the first author.

d) The first three sentences of the abstract must be moved to the introduction section since they are not for the abstract. After moving these sentences to the abstract, the authors should include citations to each topic mentioned by them. It should be noted that LPGs are also used in fiber lasers and amplifiers as stop-band optical in-line filters (not band-pass one as the authors claim; see line 4 in the abstract); the reference of the use of LPG in fiber laser as a stop-band filter can be https://doi.org/10.1063/1.4867888.

e) For long-period grating the author used two abbreviations: LPG and LPFG. They should select only one abbreviation.

f) Some commas are strange: see lines 8,16,17,21 on page 2, etc.

g) Section 2: the authors claim that “LPGs are usually formed in the media of optical rectangular waveguides”, but that is not true. LPGs are usually formed in cylindrical waveguides based on step-index fibers.

h) The same section: the authors claim that “the core and cladding refractive index difference ranges from 0.004 to 0.01”. Since NA varies from 0.06 (LMA fibers) to about 0.27 (fibers with small core), NA varies from 0.001 to 0.02 or slightly more. It should be reflected.

i) The same section: the authors should include the formula for the calculation of V and explain that the single-mode regime is observed when V < 2.405; otherwise, the regime is multi-mode. The unique propagation mode in the case of V < 2.405 is LP01. It should be mentioned.

j) The paragraph below Fig. 1 should include references on the matter.

k) Page 4: The authors should include references from which the formulae (1) and (2) were taken.

l) The sentence above Fig.5: “The LPFG is formed by controlling the above parameters”. LPG cannot be formed by controlling. It can be used, for example, for control of something.

m) Fig. 5(b) is not necessary. The grating period Lambda can be easily shown in Fig. 1(a). The same for Fig. 6.

n) The first sentence on page 5 repeats what was already discussed on page 2.

o) Page 5: reference on the LPG degradation.

p) Fig. 8: the authors should be sure that all applications are supported by references.

r) Table 1: This table is not necessary to present. All types of sensors are shown in Fig. 8. FBGs of Type 1 and Type 2 may be indicated in this Figure by different colors.

s) Page 9: The authors should include references on applications of LPGs in airspace and submarine.

Comments on the Quality of English Language

The English grammar should be strongly improved. It isn't very easy to understand some places. 

Author Response

Response to the comments

Reviewer #4:

The authors of the manuscript (MS) “Mechanically induced long-period fiber gratings and applications” review the fabrication methods of mechanically induced long-period gratings (MLPGs) and their applications. The MS can probably be suitable for publication in MDPI Photonics after careful major revision. There are so many authors but the MS is poorly prepared. My comments on the matter are given below.

Thank you for your positive comments on our study and valuable suggestions to improve the quality of our manuscript.

  1. a) The English grammar should be strongly improved. It isn't very easy to understand some places.

Thank you for your suggestion. We have made efforts to express our content as clearly as possible, making the manuscript more concise and understandable.

  1. a) To confirm that the topic of the manuscript (MS) is of increased interest to the reader, authors should include references to articles published recently, within the last two years (2022 and 2023). They must also demonstrate that the topic of the MS is important for the journal (MDPI Photonics) by showing references to the papers published in this journal. In the current MS version, there are only 6 references (9%) to the papers on the topic published during 2022-2023 and only 2 papers (3%) published in MDPI Photonics.

We thank you for your suggestion. We strive to ensure that the referenced literature supports our viewpoints and conclusions. We believe that the literature we have cited adequately covers the important research findings in the relevant field.

  1. b) Ref. 31 does not include the journal title.

Thank you for your comment. We have added the journal title of the Ref.31 in the manuscript.

  1. c) Ref. 37 does not include the first name (at least the first letter of the first name) of the first author.

Thanks for the comment. We have carefully reviewed our manuscript and ensured that the first name of the first author is included in the Ref.37.

  1. d) The first three sentences of the abstract must be moved to the introduction section since they are not for the abstract. After moving these sentences to the abstract, the authors should include citations to each topic mentioned by them. It should be noted that LPGs are also used in fiber lasers and amplifiers as stop-band optical in-line filters (not band-pass one as the authors claim; see line 4 in the abstract); the reference of the use of LPG in fiber laser as a stop-band filter can be https://doi.org/10.1063/1.4867888.

Thank you for your suggestion. I believe it is necessary to start discussing the historical development of LPFG in the abstract. LPFGs can be used as bandpass filter. See the reference of [Opt. Lett., 2010, 35(7): 1061-1063]

  1. e) For long-period grating the author used two abbreviations: LPG and LPFG. They should select only one abbreviation.

Thanks for the suggestion. The full name of LPG is long period grating. It includes long period fiber grating (LPFG) and long period waveguide grating (LPWG), they are different.

  1. f) Some commas are strange: see lines 8,16,17,21 on page 2, etc.

We thank you for your comment. We have fixed the misuse of commas.

  1. g) Section 2: the authors claim that “LPGs are usually formed in the media of optical rectangular waveguides”, but that is not true. LPGs are usually formed in cylindrical waveguides based on step-index fibers.

Thank you for your comment. In fact, LPFGs (Long Period Fiber Gratings) are usually formed in cylindrical waveguides based on step-index fibers, rather than LPGs (Long Period Gratings). We have made the necessary corrections in the manuscript to ensure accuracy.

  1. h) The same section: the authors claim that “the core and cladding refractive index difference ranges from 0.004 to 0.01”. Since NA varies from 0.06 (LMA fibers) to about 0.27 (fibers with small core), NA varies from 0.001 to 0.02 or slightly more. It should be reflected.

Thanks for the suggestion. The refractive index difference mentioned in our paper is more directed towards traditional fibers.

  1. i) The same section: the authors should include the formula for the calculation of V and explain that the single-mode regime is observed when V < 2.405; otherwise, the regime is multi-mode. The unique propagation mode in the case of V < 2.405 is LP01. It should be mentioned.

We thank the constructive suggestion. We have clarified this point in the second section of the manuscript.

  1. j) The paragraph below Fig. 1 should include references on the matter.

Thanks for your suggestion. We considered that SMF is well known, so it can be understood even without a reference.

  1. k) Page 4: The authors should include references from which the formulae (1) and (2) were taken.

Thank you for your suggestion. We believed that equations (1) and (2) are widely known, so we did not cite specific references.

  1. l) The sentence above Fig.5: “The LPFG is formed by controlling the above parameters”. LPG cannot be formed by controlling. It can be used, for example, for control of something.

We thank you for your comment. The resonant dips of LPFG are controlled the above parameters.

  1. m) Fig. 5(b) is not necessary. The grating period Lambda can be easily shown in Fig. 1(a). The same for Fig. 6.

Thank you for your suggestion. Including figures 5(b) and 6(b) here is for the better view.

  1. n) The first sentence on page 5 repeats what was already discussed on page 2.

Thank you for your comment. Regarding your observation about the repetition of the first sentence on page 5, which was already discussed on page 2, we have fixed this issue in our manuscript.

  1. o) Page 5: reference on the LPG degradation.

We thank you for your comment. It is because the degradation that make the mechanically induced LPFG to be reconfigurable.

  1. p) Fig. 8: the authors should be sure that all applications are supported by references.

Thanks for your suggestion. We are sure that all the applications are supported by references and correspond to their respective references.

  1. r) Table 1: This table is not necessary to present. All types of sensors are shown in Fig. 8. FBGs of Type 1 and Type 2 may be indicated in this Figure by different colors.

We thank you for your suggestion. It is straightforward to use Table 1. In this manuscript, We do not talk about FBG. We have indicated the two types of mechanically induced LPFG using different colors in Table 2.

  1. s) Page 9: The authors should include references on applications of LPGs in airspace and submarine.

Thank you for your suggestion. To our best effort, we have not found the references yet.

Round 2

Reviewer 4 Report

Comments and Suggestions for Authors

The authors of the manuscript (MS) “Mechanically induced long-period fiber gratings and applications” review the fabrication methods of mechanically induced long-period gratings (MLPGs) and their applications. The MS is suitable for publication in MDPI Photonics as it is.